# Feasibility of Antimicrobial Stewardship (AMS) in Critical Care Settings: A Multidisciplinary Approach Strategy

**DOI:** 10.3390/medsci6020040

**Published:** 2018-05-25

**Authors:** Tamas Tiszai-Szucs, Claire Mac Sweeney, Joseph Keaveny, Fernando A. Bozza, Zieta O. Hagan, Ignacio Martin-Loeches

**Affiliations:** 1Department of Anaesthesia and Critical Care Medicine, St James’s Hospital, P.O. Box 580 Dublin 8, Ireland; tiszai_szucs@yahoo.com (T.T.-S.); macsweec@gmail.com (C.M.S.); jakeaveny@gmail.com (J.K.); ZOHagan@stjames.ie (Z.O.H.); 2Fundacao Oswaldo Cruz, Rio de Janeiro, RJ 37903, Brazil; bozza.fernando@gmail.com; 3Multidisciplinary Intensive Care Research Organization (MICRO), St James’s Hospital, P.O. Box 580 Dublin 8, Ireland; 4Trinity Centre for Health Sciences, P.O. Box 580 Dublin 8, Ireland; 5CIBER enfermedades respiratorias, 28029 Madrid, Spain

**Keywords:** resistance, antimicrobial stewardship (AMS), pneumonia, sepsis, ventilator-associated pneumonia (VAP), intensive care unit (ICU), multidrug resistance (MDR), stewardship, antibiotics

## Abstract

Antimicrobial resistance is escalating and triggers clinical decision-making challenges when treating infections in patients admitted to intensive care units (ICU). Antimicrobial stewardship (AMS) may help combat this problem, but it can be difficult to implement in critical care settings. The implementation of multidisciplinary AMS in ICUs could be more challenging than what is currently suggested in the literature. Our main goal was to analyze the reduction in duration of treatment (DOT) for the most commonly used antibacterial and antifungal agents during the first six months of 2014, and during the same period two years later (2016). A total of 426 and 424 patient encounters, respectively, were documented and collected from the intensive care unit’s electronic patient record system. Daily multidisciplinary ward rounds were conducted for approximately 30–40 min, with the goal of optimizing antimicrobial therapy in order to analyze the feasibility of implementing AMS. The only antimicrobial agent which showed a significant reduction in the number of prescriptions and in the duration of treatment during the second audit was vancomycin, while linezolid showed an increase in the number of prescriptions with no significant prolongation of the duration of treatment. A trend of reduction was also seen in the DOT for co-amoxiclavulanate and in the number of prescriptions of anidulafungin without any corresponding increases being observed for other broad-spectrum anti-infective agents (*p*-values of 0.07 and 0.05, respectively).

## 1. Introduction

Antimicrobial resistance is escalating worldwide, resulting in higher morbidity rates, mortality rates, and cost [1]. When treating critically ill patients, intensivists must consider not only the risks associated with delays in initiating antimicrobial therapy, but also the potential selection for and development of multidrug-resistant organisms (MDROs) [2]. Rationalization of the use of antibiotics is a cornerstone of slowing the development and spread of MDROs [3].

Antimicrobial stewardship (AMS) can optimize the treatment of infections, while reducing the adverse effects of excessive and unnecessary antimicrobial use, such as the development of MDROs [4]. A consensus on the required components of AMS is yet to be reached, and its impact on mortality remains unproven [5]; however, its beneficial effects in terms of consumption, cost, and resistance rates have been confirmed [6]. AMS entails frequent reassessment and de-escalation of antibiotic therapies, and a reduction in the duration of their administration, while avoiding multi-agent protocols where possible, and using rapid diagnostic tools to rule out infection, thereby avoiding unnecessary antimicrobial treatment [7].

Bacterial resistance profiles vary widely internationally, within Europe, within communities, and within various hospital settings. The European Antimicrobial Resistance Surveillance Network (EARS-Net) collects antimicrobial susceptibility data from European Union (EU) member states. Our data was collected in European and Irish contexts, based on their November 2017 publication [8]. While submission of microbiological data to EARS-Net is compulsory, individual centers are responsible for ascertaining antimicrobial prescription patterns and how they might relate to resistance. Our goal was to assess the feasibility of implementing multidisciplinary AMS in critical care settings, as it could be more challenging than what is currently suggested by literature. We compared the number of prescriptions and the durations of treatment (DOT) for the most commonly used antibacterial and antifungal agents in intensive care units (ICUs) across two periods of six months.

## 2. Methods

### 2.1. Patient Recruitment and Data Collection

The presented longitudinal study was based on two audits performed in the general ICU of St. James’s Hospital, Dublin, which is the largest hospital in the Republic of Ireland. Local antibiotic consumption in response to the most clinically significant bacterial and fungal pathogens, in terms of incidence and susceptibility, was analyzed across two audits carried out during two equal six-month periods (1 January to 31 July 2014, and 1 January to 31 July 2016). Following analysis of results from the first audit, a multidisciplinary-AMS campaign was initiated by means of local joint meetings, the education of trainees, and daily supervision of clinical practice, with improvement of the AMS carried out with the assistance of the microbiology department of the hospital. The study was conducted in accordance with the Declaration of Helsinki, and protocols were waived by the Research Ethics Committee at St James’s Hospital, based on the auditorial nature of the study.

We collected susceptibility results for the above-mentioned pathogens in collaboration with the microbiology department of the hospital, and separately analyzed the susceptibility results of isolates instead of on a case-by-case basis. During both study periods, we audited every admission to the general intensive care unit, choosing to include surgical and medical patients, but not patients who were initially admitted to ICUs in the burns and cardiothoracic units of the hospital. Each admission was considered to be a separate patient encounter, including patients who were readmitted to the ICU. We collected demographic data (age, gender, and length of stay) and study-specific variables, including prevalence of bacterial and fungal pathogens, susceptibility results for certain bacteria, antimicrobial use, and length of treatment [9].

Demographic and clinical data, including length of stay, specialty (medical or surgical) under which patients were admitted, presence of clinical pneumonia, antimicrobial use, and duration of treatment were collected from the ICU’s electronic record system (IntelliVue Clinical Information Portfolio). The microbiology laboratory provided microbiological results, including pathogen occurrence and susceptibility results. The analyzed microbiological data included results from respiratory samples (including sputum, and bronchoalveolar lavage), blood cultures, and various fluid samples (either pleural or abdominal drain samples) obtained when clinical infection was suspected. Urinary line tips, swab samples, and viral screening results were excluded. Incidences of influenza were confirmed through RT-PCR. Rectal swabs were obtained for the purpose of infection control and to screen for vancomycin-resistant enterococci (VRE). The five most commonly isolated gram-negative bacteria during both study periods were reported: *Escherichia coli* (*E. coli*), *Pseudomonas aeruginosa* (*P. aeruginosa*), *Klebsiella pneumoniae* (*K. pneumoniae*), *Klebsiella oxytoca* (*K. oxytoca*), and *Enterobacter cloacae* (*E. cloacae*). 

During the six-month surveillance period, daily multidisciplinary ward rounds (including intensivists, clinical microbiologists, and pharmacists) were undertaken to implement AMS, with each round taking approximately 30–40 min. Decisions regarding antibiotic prescription, duration of treatment, and any changes to either were based on a consensus within the multidisciplinary team, with the final decision made by the attending intensivist. 

### 2.2. Primary Objective

Our main goal was to assess the feasibility of reducing the DOT of the most commonly used antibacterial and antifungal agents, by analyzing their use in the first two six-month periods of 2014 and 2016.

### 2.3. Secondary Objectives

Secondary goals included a reduction in cumulative frequency of fungi, and the four most common gram-negative bacteria defined in the first audit, and an evaluation of any decrease in antimicrobial resistance, presented as the frequency of resistant pathogens in the total number of positive isolates, calculated during both study periods.

### 2.4. Statistical Analysis

Categorical and continuous data were presented as numbers (percentages), and as medians and interquartile ranges, respectively. Each DOT (number of days) was reported with a median and an interquartile range (IQR). Categorical variables were compared using Fisher’s exact test. Quantitative continuous variables were compared using the Mann–Whitney test. A two-sided *p*-value ≤ 0.05 was considered statistically significant. Data analysis was performed using the Statistical Package of the Social Sciences (SPSS) program for Windows 22.0 (SPSS, Chicago, IL, USA).

## 3. Results

### 3.1. Demographic Data

During the first six months of 2014 and 2016, 426 and 424 patient encounters, respectively, were documented and collected from the intensive care unit’s electronic patient record system. The median age of patients was 64 years-old (IQR: 50–73 years-old) in 2014, and 60 years-old (IQR: 47–72.25 years old) in 2016 (*p* = not significant, n.s). In the first group, there were 237 male patients (55.63%), while in the second group, 259 participants (60.80%) were men (*p* = n.s). Clinical pneumonia was diagnosed in 219 cases (51.41%) in the first group, and in 142 cases (33.49%) in the second group (*p* = 0.0001). The median length of stay was 3.87 (IQR: 1.75–8.12 days) and 2.71 (IQR: 1.4–5.96 days) days (*p* = 0.057) in 2014 and 2016, respectively.

### 3.2. Microbiological Results

This section details the occurrence and susceptibility results of the most common gram-negative bacteria to further analyze the consumption of broad-spectrum antibiotics. Influenza occurred significantly higher (*p* = 0.006) in the second period (16/424 vs. 4/426). In 2014, the most common gram-negative bacteria in order of the number of occurrences were *E. coli*, *P. aeruginosa*, *K. pneumoniae*, *K. oxytoca*, and *E. cloacae*. All of these pathogens were significantly more frequently isolated in 2014 than in 2016, predominantly from respiratory samples.

*E. coli* demonstrated resistance in more than 20% of cases in 2014 and 2016 (amoxicillin—83.33% and 66.67%, respectively; co-amoxiclavulanate—58.33% and 22.22%, respectively; and ciprofloxacin—33.33% and 33.33%, respectively), and proved to be mostly sensitive to meropenem, piperacillin/tazobactam, amikacin, gentamicin, ceftazidime, cefotaxime, tigecycline, and ertapenem in both audits. For the other selected gram-negative pathogens, we only calculated resistance rates for the first audit due to a low number of isolates in 2016. *P. aeruginosa* was resistant in 29.41% of cases to both piperacillin/tazobactam and ceftazidime in 2014, while remaining predominantly sensitive to meropenem, amikacin, gentamicin, and ciprofloxacin in the first audit, with the second audit showing a similar pattern. *K. pneumoniae* showed high resistance rates to amoxicillin (100%) and piperacillin/tazobactam (25%), while remaining mostly sensitive to meropenem, amikacin, ciprofloxacin, gentamicin, ceftazidime, cefotaxime, ertapenem, and tigecycline in 2014. *K. oxytoca* was resistant to amoxicillin in 100% of cases, and resistant to both co-amoxiclavulanate and piperacillin/tazobactam in 54.55% of cases in the first audit. Additionally, *K. oxytoca* was resistant, in 20% of cases, to meropenem, amikacin, ciprofloxacin, gentamicin, ceftazidime, cefotaxime, tigecycline, ertapenem, and aztreonam. *E. cloacae* was resistant 20% to co-amoxiclavulanate (100%), piperacillin/tazobactam (20%), ceftazidime (20%), cefotaxime (20%), and aztreonam (20%), while being sensitive to meropenem, amikacin, ciprofloxacin, gentamicin, tigecycline, and ertapenem in more than 80% of isolates in 2014.

Among the Gram-positive pathogens, we evaluated the frequencies of *Staphylococcus aureus* (*S. aureus*), including methicillin-resistant and methicillin-sensitive cases, and *Enterococcus faecium* (*E. faecium*), including vancomycin-resistant and vancomycin-sensitive cases, comparing them to the total number of microbiological samples taken. *S. aureus* was significantly more often isolated in the first audit (in 36 microbiological samples), and was predominantly found in respiratory samples, while *S. aureus* was only isolated in one sample in 2016. Methicillin-resistant *S. aureus* (MRSA) was found in 16.67% of cases in 2014, while there was no MRSA isolated among the microbiological samples analyzed in 2016. *E. faecium* was isolated from 38 and 32 microbiological samples in 2014 and 2016, respectively (*p* = not significant), and was isolated from faecal samples significantly more frequently in the first audit than in the second. VRE were found in 90.48% of cases in 2014, and in 93.45% of cases in 2016 (*p* = n.s). Further details are displayed in Table 1.

*Candida* sp. and *Aspergillus* sp. were the most common fungal pathogens in both audits. *Candida* sp. was isolated in 158 (13.00%) and 19 (1.51%) isolates in the first and second audits, respectively (*p* = 0.0001), while *Aspergillus* sp. were found in 12 isolates in both 2014 and 2016. *Candida* sp. were predominantly and significantly more frequently isolated from respiratory samples in the first audit, reflecting a high rate of colonization; however, candidemia was not significantly different between study periods (Table 1). Respiratory samples were the only source of *Aspergillus* during both study periods.

### 3.3. Antibacterial Consumption and Duration of Treatment

In order to quantify antimicrobial consumption, we calculated the DOT for antibiotics that were used in more than 20% of cases. Data of antibiotic consumption and duration comparisons between study periods are displayed in Table 2. In brief, the number of prescriptions and the duration of vancomycin treatment were both reduced in the second audit despite the fact that the number of prescriptions of linezolid was higher in the second period without its duration of treatment being significantly prolonged.

## 4. Discussion

Our main goal was to analyze the reduction in DOT for the most commonly used antibacterial and antifungal agents. In the first audit, a discrepancy was noted between occurrences of *S. aureus*, more specifically MRSA, and vancomycin use, which was accompanied by a relatively high rate of vancomycin resistance. One of the major findings was that vancomycin, the second most frequently used antibiotic, was overprescribed; hence, we were particularly focused on limiting vancomycin use. Antimicrobial stewardship (AMS) applied between 2014 and 2016 was associated with a significant reduction in the absolute number and the duration of vancomycin prescriptions, as well as the number of occurrences of *S. aureus* and MRSA, while VRE rates remained unchanged.

In order to implement multidisciplinary AMS, we put in place a multistep process including audits and the implementation of microbiologist-intensivist-pharmacist meetings, with the main goal being to reduce the DOT for the most commonly used antibacterial and antifungal agents. Following the first audit, we were especially concerned by the very high rate of vancomycin consumption, but we wanted to develop a broad approach, including all antibiotic and antifungal classes [10,11]. We de-escalated MRSA regimens, in terms of the number of prescriptions of, and the durations of treatment with vancomycin. While the number of prescriptions was higher in the second period, the duration of treatment with linezolid was not significantly prolonged. The gram-negative pathogens that we analyzed were predominantly isolated in respiratory samples and showed a significantly lower level of occurrence in the second period. This was likely related to our other finding that significantly more patients were diagnosed with pneumonia in the first audit. In 2014, there were also significantly more surgical admissions, with a trend of longer ICU stays. *E. coli* and *Klebsiella* sp. were mostly sensitive to meropenem, cephalosporins, and aminoglycosides in both study groups, and were less commonly sensitive to penicillin derivatives or ciprofloxacin. Furthermore, *Pseudomonas* sp. were mainly sensitive to aminoglycosides and ciprofloxacin in both study groups. Finally, *Enterobacter* sp. were sensitive to meropenem, aminoglycosides, and ciprofloxacin in more than 80% of isolates during both study periods. The above findings showed a favorable susceptibility pattern against cephalosporins and aminoglycosides in most Gram-negative pathogens, in contrast to the escalating resistance reported internationally [3,9]. Thus, we concluded that the most commonly used penicillin derivatives, including piperacillin/tazobactam, co-amoxiclavulanate, and meropenem, were likely overused in a similarly high proportion of cases in both audits. Of note, all the above-listed Gram-negative bacteria showed low antimicrobial resistance to ertapenem and tigecycline, demonstrating its availability as a last resort in treating multidrug-resistant gram-negative infections [12].

Our study had some limitations. Firstly, the most important limitation was that the median number of respiratory and blood samples for each patient was not collected to analyze the impact of AMS using the standard practice of sampling; moreover, this was a non-randomized retrospective longitudinal study. Our demographic data had some significant differences between the two audit periods, including a higher incidence of clinically-suspected pneumonia; however, colony-forming units (CFU) were not used as a threshold in the evaluation of respiratory tract samples. We did not include outcome measures, other than length of stay, limiting our ability to compare outcome effects of any changes in antibacterial consumption. Furthermore, samples were taken when clinically indicated, and there was no discrimination with regards to timing of admission. We also did not include indications of antibiotic prescriptions, and reasons for escalation of therapy, limiting our attempts to rationalize antimicrobial surveillance. Finally, the true number of infected (hospital-acquired pneumonia vs. ventilator-associated pneumonia, early/late ratio) or just colonized patients was not available. Because pneumonia diagnosis was mainly clinical, both the number of diagnoses and the level of sampling were dramatically reduced, which impacted the number of episodes treated. Possible differences in populations (e.g., SAPS II and infection at admission) and clinical management (e.g., infection control and surveillance cultures) of ICU patients between 2014 and 2016, beyond those due to AMS, may have been significant and interfered with the results observed. However, no changes in clinical practice, outside of those due to AMS, including surveillance cultures, were recently implemented in our ICU.

In summary, our study highlighted that regular auditing of the local pathogen populations and of antimicrobial consumption is necessary in intensive care settings [5,13]. In our two-phased audit, we analyzed the feasibility of limiting the number of prescriptions of broad-spectrum antibiotics. A significant reduction in the number of prescriptions and the duration of treatment was only observed for vancomycin during the second audit despite the fact that the number of prescriptions of linezolid was higher without its duration of treatment being significantly prolonged. A trend of reduction was also seen in the DOT for co-amoxiclavulanate and in the number of prescriptions of anidulafungin, without any corresponding increases being observed for other antibiotics (*p*-values of 0.07 and 0.05, respectively). 

## Figures and Tables

**Table 1 medsci-06-00040-t001:** Isolates collected grouped by their absolute numbers and their source of isolation.

**Gram-positive bacteria**					
*Staphylococcus aureus*	Audit 1 (n = 36)		Audit 2 (n = 1)		*p*-value
Total number	36	2.96%	1	0.08%	0.0001
Respiratory culture	26	9.67%	0	0.00%	0.0001
Blood cultures	6	0.84%	1	0.12%	0.0538
Fluid samples *	4	3.57%	0	0.00%	0.0075
*Enterococcus faecium*	Audit 1 (n = 38)		Audit 2 (n = 32)		*p-*value
Total number	38	3.13%	32	2.54%	0.398
Respiratory culture	1	0.37%	0	0.00%	1
Blood cultures	10	1.40%	4	0.48%	0.0644
Fluid samples *	8	7.14%	9	4.89%	0.4472
Faeces	23	20.54%	21	11.41%	0.0426
**Gram-negative bacteria**					
*Escherichia coli*	Audit 1 (n = 25)		Audit 2 (n = 9)		*p*-value
Total number	25	2.06%	9	0.71%	0.0051
Respiratory culture	11	4.09%	0	0.00%	0.0213
Blood cultures	3	0.42%	4	0.48%	0.3642
Fluid samples *	11	9.82%	5	2.72%	0.0146
*Pseudomonas* sp.	Audit 1 (n = 17)		Audit 2 (n = 2)		*p*-value
Total number	17	1.40%	2	0.16%	0.001
Respiratory culture	17	6.32%	0	0.00%	n.s.
Blood cultures	0	0.00%	0	0.00%	n.s.
Fluid samples *	0	0.00%	2	1.09%	n.s.
*Klebsiella pneumoniae*	Audit 1 (n = 12)		Audit 2 (n = 1)		*p*-value
Total number	12	0.99%	1	0.08%	0.001
Respiratory culture	10	3.72%	0	0.00%	n.s.
Blood cultures	0	0.00%	0	0.00%	n.s.
Fluid samples *	2	1.79%	1	0.54%	n.s.
*Klebsiella oxytoca*	Audit 1 (n = 11)		Audit 2 (n = 0)		*p*-value
Total number	11	0.91%	N/A	N/A	
Respiratory culture	11	4.09%	N/A	N/A	n.s.
Blood cultures	0	0.00%	N/A	N/A	n.s.
Fluid samples *	0	0.00%	N/A	N/A	n.s.
*Enterobacter cloacae*	Audit 1 (n = 13)		Audit 2 (n = 2)		*p*-value
Total number	10	0.82%	2	0.16%	0.001
Respiratory culture	7	2.60%	0	0.00%	n.s.
Blood cultures	0	0.00%	1	0.12%	n.s.
Fluid samples *	3	2.68%	1	0.54%	n.s.
**Fungal pathogens**					
*Candida* sp.	Audit 1 (n = 158)		Audit 2 (n = 19)		*p*-value
Total number	158	13.00%	19	1.51%	0.0001
Respiratory culture	139	51.67%	2	1.71%	0.0001
Blood cultures	3	0.42%	5	0.60%	0.732
Fluid samples *	13	11.61%	12	6.52%	0.1365
*Aspergillus* sp.	Audit 1 (n = 12)		Audit 2 (n = 12)		*p*-value
Total number	12	0.99%	12	0.95%	1
Respiratory culture	12	4.46%	12	10.26%	0.0387

* Either pleural or abdominal drain samples. n.s = non significant.

**Table 2 medsci-06-00040-t002:** Antibiotic consumption and duration comparisons between study periods.

Antibacterial Consumption					
	Audit 1 (n = 426)		Audit 2 (n = 424)		
**Antibiotic (in >10% of cases)**	*Number (No.) of cases*	*Percentage*	*No. of cases*	*Percentage*	*p-*value
Piperacillin/tazobactam	218	51.17%	219	51.65%	0.891
Vancomycin	174	40.85%	127	29.95%	0.001
Co-amoxiclavulanate	146	34.27%	124	29.25%	0.122
Metronidazole	137	32.16%	115	27.12%	0.115
Meropenem	131	30.75%	115	27.12%	0.257
Gentamicin	106	24.88%	108	25.47%	0.875
Ciprofloxacin	75	17.61%	47	11.08%	0.008
Clarithromycin	62	14.55%	57	13.44%	0.693
Linezolid	53	12.44%	78	18.40%	0.018
**Duration of treatment**					
	Audit 1 (n = 426)		Audit 2 (n = 424)		
**Antibiotic (in > 10% of cases)**	*Median (days)*	*Interquartile range (IQR) (days)*	*Median (days)*	*IQR (days)*	*p*-value
Piperacillin/tazobactam	3.67	(2–6.67)	3.33	(2–6)	0.249
Vancomycin	4	(2.22–7.02)	2.50	(1.5–5.13)	0.001
Co-amoxiclavulanate	2.33	(1.33–3)	1.67	(1.33–2.67)	0.071
Metronidazole	3.33	(1.96–7.51)	2.67	(1.67–4.33)	0.107
Meropenem	5	(2.67–8.03)	6.04	(2.83–10.32)	0.348
Gentamicin	1	(1–3)	1.95	(1–3)	0.395
Ciprofloxacin	3.5	(2–6.66)	3.00	(2–6.42)	0.670
Clarithromycin	3.5	(2.5–5.5)	3.00	(2–4.5)	0.127
Linezolid	3.5	(2–6.5)	3.50	(2–7.38)	0.875
**Antifungal consumption**					
	Audit 1 (n = 426)		Audit 2 (n = 424)		
***Antifungal***	*No. of cases*	*Percentage*	*No. of cases*	*Percentage*	*p*-value
Anidulafungin	72	16.90%	51	12.03%	0.051
Fluconazole	42	9.86%	42	9.91%	1
Amphotericin B (Liposomal)	27	6.34%	16	3.77%	0.117
Caspofungin	9	2.11%	10	2.36%	0.821
Voriconazole	7	1.64%	8	1.89%	0.802
**Duration of treatment**					
	Audit 1 (n = 426)		Audit 2 (n = 424)		
***Antifungal***	*Median (days)*	*IQR (days)*	*Median (days)*	*IQR (days)*	*p*-value
Anidulafungin	4.00	(2–9)	4.00	(2.27–6)	0.79
Fluconazole	3.00	(2–6.75)	4.95	(2.5–6)	0.25
Amphotericin B (Liposomal)	5.00	(2–12.08)	5.46	(2.25–9.56)	0.93
Caspofungin	3.00	(1.92–4.17)	4.00	(2–5)	0.84
Voriconazole	5.92	(2.75–12.52)	6.00	(3.88–9.70)	0.95

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
