# Peer review of "Feasibility of Antimicrobial Stewardship (AMS) in Critical Care Settings: A Multidisciplinary Approach Strategy"

_medsci, 2018, doi:10.3390/medsci6020040_

Round 1

Reviewer 1 Report

In the present manuscript (medsci-286003) the authors try to evaluate the effectiveness of a multidisciplinary antimicrobial stewardship program through a two-stage retrospective audit series. They have found a decrease in the occurrences of S. aureus and MRSA, as well as in vancomycin prescriptions. The idea of the study is interesting. However, there are several concerns that have to be commented.

Major concerns:

In my humble opinion, the most important concern with the work is that, in the way it has been written, the aim is not clear. There is a lot of data that is dispensable as it does not provide useful information, and, on the other hand, there is missing information regarding the objectives of the study. For instance, beginning with the title, it talks about “feasibility” of a program. But there is no information explaining if the program is feasible or not, only information explaining microbiological isolates and antibiotic usage in the two periods. Furthermore, in the abstract, the authors talk about “effectiveness” of the program, and not feasibility. Afterward, in the introduction, they write “Our main goal when conducting this study was to implement a multidisciplinary audit for 
antimicrobial stewardship in order to assess the feasibility of such programs in contemporary critical care practice in Ireland”.
And even they say that their hypothesis was that multidisciplinary AMS campaign is a challenging program that is more difficult to implement than it is suggested by the current literature. Moreover, they want to see this effectiveness through antimicrobial consumption, but also microbiological isolates are included. And finally, they say that they want to focus on vancomycin, but also other data regarding gram-negative bacteria and other antibiotics are provided. These last data only shows the picture of the isolates in a concrete ICU, not the effect of the ASP, and I do not think it is interesting for most of the readers. Therefore, I would suggest clarifying these points, beginning with the title and establishing a clear aim and objective, giving the data regarding this aim and objective.

Another major concern is that the used methods (as well as the way they have been explained) can be improved.  First, there is no information about the antimicrobial stewardship. In the abstract, it is not explained that it was done between the two audits, and in the methods section it is only explained that there were meetings, but the real content of the program is not specified. Depending on what issues are emphasized in the ASP, they could or not explain some results. For instance, it seems, following authors comments, that they were especially worried about vancomycin consume. If this point is emphasized in the ASP, it could explain the reduction in vancomycin consume. A brief description is therefore needed.

Moreover, as the audit periods are relatively short, it should be explained if some clinical relevant issues happened during that period that could explain some results. For instance, there was more pneumonia in the first period, with more S. aureus isolates. Were there more patients with influenza admitted? As influenza can complicate with S. aureus pneumonia, it could explain some results, even the reduction in the consumption of vancomycin. If not, why there were more patients with pneumonia?. And the reduction in vancomycin, was due to an increase in linezolid or other antibiotics against gram-positive bacteria (ceftaroline, for instance). And how can the authors explain the huge differences in Candida isolates?. Even more important: were the microorganisms isolated at admission or after some days in the ICU. Because if they were at admission, changes are not due to the ASP, meanwhile if they were isolated during ICU stay, they could.

Finally, the conclusions are not based on the results of the study.

Minor concerns:

1.     The way the manuscript is written is somehow disorganized and unclear to the reader. I suggest rewrite some parts and change the order to make it clear. For instance, some brackets are lacking in the results section or even some numbers (line 615), or an “a” should not appear in K. a. pneumoniae (line 637), repetition of “high-dose” in line 727, among others.

2.     The antimicrobial stewardship program sometimes is expressed as ASP and others as AMS. It should be homogenized.

3.     Line 639-645. This paragraph should appear in the methods section, not in results.

4.     Tables are not referred in the text.

Author Response

In the present manuscript (medsci-286003) the authors try to evaluate the effectiveness of a multidisciplinary antimicrobial stewardship program through a two-stage retrospective audit series. They have found a decrease in the occurrences of S. aureus and MRSA, as well as in vancomycin prescriptions. The idea of the study is interesting. However, there are several concerns that have to be commented.

Major concerns:

In my humble opinion, the most important concern with the work is that, in the way it has been written, the aim is not clear. There is a lot of data that is dispensable as it does not provide useful information, and, on the other hand, there is missing information regarding the objectives of the study. For instance, beginning with the title, it talks about “feasibility” of a program. But there is no information explaining if the program is feasible or not, only information explaining microbiological isolates and antibiotic usage in the two periods. Furthermore, in the abstract, the authors talk about “effectiveness” of the program, and not feasibility. Afterward, in the introduction, they write “Our main goal when conducting this study was to implement a multidisciplinary audit for antimicrobial stewardship in order to assess the feasibility of such programs in contemporary critical care practice in Ireland”.And even they say that their hypothesis was that multidisciplinary AMS campaign is a challenging program that is more difficult to implement than it is suggested by the current literature. Moreover, they want to see this effectiveness through antimicrobial consumption, but also microbiological isolates are included. And finally, they say that they want to focus on vancomycin, but also other data regarding gram-negative bacteria and other antibiotics are provided. These last data only shows the picture of the isolates in a concrete ICU, not the effect of the ASP, and I do not think it is interesting for most of the readers. Therefore, I would suggest clarifying these points, beginning with the title and establishing a clear aim and objective, giving the data regarding this aim and objective.

Response: We think the reviewer is right, as the other reviewer also pointed it out. Our main goal was to analyze the reduction of duration of treatment (DOT) of the most commonly used antibacterial and antifungal agents. We have also incorporate limitations about the microbiology/ecology and antibiotic consumption rates rather than clinical data that represent a major limitation in our study. We have amended and tone down the discussion, harmonized the goals throughout the manuscript. We have done a major revision in the current manuscript that we hope it will help to understand better the aim.

Another major concern is that the used methods (as well as the way they have been explained) can be improved.  First, there is no information about the antimicrobial stewardship. In the abstract, it is not explained that it was done between the two audits, and in the methods section it is only explained that there were meetings, but the real content of the program is not specified. Depending on what issues are emphasized in the ASP, they could or not explain some results. For instance, it seems, following authors comments, that they were especially worried about vancomycin consume. If this point is emphasized in the ASP, it could explain the reduction in vancomycin consume. A brief description is therefore needed.

Moreover, as the audit periods are relatively short, it should be explained if some clinical relevant issues happened during that period that could explain some results. For instance, there was more pneumonia in the first period, with more S. aureus isolates. Were there more patients with influenza admitted? As influenza can complicate with S. aureus pneumonia, it could explain some results, even the reduction in the consumption of vancomycin. If not, why there were more patients with pneumonia?. And the reduction in vancomycin, was due to an increase in linezolid or other antibiotics against gram-positive bacteria (ceftaroline, for instance). And how can the authors explain the huge differences in Candida isolates?. Even more important: were the microorganisms isolated at admission or after some days in the ICU. Because if they were at admission, changes are not due to the ASP, meanwhile if they were isolated during ICU stay, they could.

Response: All of these points have also been raised by the other reviewer and we agree with all the criticism. In order to implement a multidisciplinary AMS, we put in place a multi-step process including audit, implementation of microbiology-intensivist-Pharmacist meetings with the main goal of reducing of duration of treatment (DOT) of the most commonly used antibacterial and antifungal agents. We were especially concerned after the first audit in the very high vancomycin consumption but we wanted to develop a broad approach including all antibiotic and antifungal classes. This paragraph has been included in the discussion to make the manuscript more focused to the readers.

Influenza was not included but based on the reviewer’s comment we have searched this data and found that influenza was more frequent in 2016 than in 2014 during the study period of analysis “Influenza was significantly higher (p 0.006) in the second period (16/424 vs. 4/426)”. This information has been now included. We agree with the criticism of linezolid however we could de-escalate more often MRSA regimens in prescription and duration with Vancomycin and while prescription was higher in the second period for Linezolid, the duration of the treatment was not significantly prolonged. We have also incorporated this finding. Ceftaroline was not available in our hospital when the study was conducted. Regarding Candida isolates, whilst we can demonstrate that there was a change in sampling practice, the reduction in candida colonization might represent a better implementation of less sample to be sent with lower clinical suspicion. One of the main goals of AMS is the lower sampling that both save cost and implement cost to daily clinical practice. We have to agree that we can´t demonstrate this finding with data but could be a hypothesis generating answer to reviewer’s question. Finally, the samples were taken when clinically indicated and there was no discrimination of timing to admission. We have included this point as a limitation.

Finally, the conclusions are not based on the results of the study.

Response: reviewer is right with this criticism and conclusions have been amended and as a final paragraph we have highlight the need of auditing local pathogen flora and the main finding mainly in gram positives. “In summary, our study highlighted that regular auditing of the local pathogen flora and antimicrobial consumption is necessary in intensive care settings. In our two-phased audit, we analysed the feasibility to limit the prescription of broad-spectrum antibiotics. As a major finding, prescription and duration of Vancomycin was reduced in the second audit despite the fact that prescription of linezolid was higher in the second period but duration was not significantly prolonged. MRSA rates were significantly reduced, while VRE rates remained unchanged”.

Minor concerns:

1.     The way the manuscript is written is somehow disorganized and unclear to the reader. I suggest rewrite some parts and change the order to make it clear. For instance, some brackets are lacking in the results section or even some numbers (line 615), or an “a” should not appear in K. a. pneumoniae (line 637), repetition of “high-dose” in line 727, among others.

Response: We agree and the manuscript has been proof read to amend the aforementioned mistakes.

2.     The antimicrobial stewardship program sometimes is expressed as ASP and others as AMS. It should be homogenized.

Response: As commented to Rev1, we have homogenized to AMS

3.     Line 639-645. This paragraph should appear in the methods section, not in results.

Response: We have done it.

4.     Tables are not referred in the text.

Response: Tables are now referred in the text.

Reviewer 2 Report

MAJOR

The author proposed an experience of antimicrobial stewardship ICU program (ASP) with relevant results in terms of antibiotic therapy prescription. The data are interesting but need some further clarification. First of all, the aim of the study should be defined and homogenized in the different sections of the manuscript. By results and discussion, the aim seems to be the evaluation of the effectiveness of the ASP in terms of antibiotic use and micro-organisms antibiotic resistance rather than a feasibility study. Unfortunately, the effects of ASP on infection incidence is unclear because data on pneumonia are incomplete. Moreover, differences in populations (e.g. SAPS II, infection at admission) and clinical management (e.g. infection control, surveillance cultures) of ICU patients, beyond ASP, between 2014 and 2016 might be significant and interfere with the results observed. More data on the population and the standard clinical practice in the 2 periods should be reported.

MINOR

ABSTRACT

-          For antimicrobial stewardship, 2 acronyms (ASP and AMS) have been used. I suggest to use just one

-          The intervention (ASP) should be better clarified in terms of activities and time

-          A conclusion should be added

INTRODUCTION

                Lines 536-538: The aim of the study should be better clarified: in the abstract and in the methods the aim seems to be the effectiveness of ASP, whereas in the title and in the introduction seems to be only feasibility.

METHODS

-          Line 545: Sensitivity (to antibiotics) might be changed to susceptibility to antibiotics.

-          Line 565: Was tracheal aspirate not considered or never performed ?

-          Line 566: please clarify ‘various fluid samples (categorised as pleural of various other drain samples)’.

-          Lines 566-568: the authors stated that they did not consider urine, rectal swab, line tip and viral samples because the study was focused on bloodstreams and respiratory infections. various fluid samples (categorised as pleural of various other drain samples) and fecal samples. Therefore, it is unclear the reason for including fecal samples in the analysis

-          Have a CFU threshold used in the evaluation of the respiratory tract samples ?

-          Lines 572-577: I suggest rephrasing the primary end point. DOT could be not a endpoint, but just a variable. The reduction of DOT may be an end-point. The goal (antibiotic consumption improvement) is unclear. More, the relationship between end-point and hypothesis is difficult to realize. In general, the hypothesis should be placed at the beginning of the paragraph (or better in the introduction)

-          Lines 579-583: As above, the secondary end-points have to be better defined.

RESULTS

-          Lines 632-633: The lines should be rephrased.

-          Lines 632-645: I suggest to move this section in the methods

-          Lines 644-646: Is the difference in susceptibility significant ?

-          Lines 654-663: I suggest to shorten these data and to use a table

-          Lines 664-669: The rate of candidemia should be reported for the 2 study periods and also the type of Candida. Data on respiratory isolation without data on invasive candida may be misleading.

-          The median (IQR) number of respiratory and blood samples for each patient should be reported. It is important to specify if the ASP changed the standard practice in terms of sampling (surveillance culture ?)

DISCUSSION

-          Lines 672-673: Again, the end-point is different to end-points defined in the abstract, introduction and methods

-          Limitations: The study is based on microbiological data and not on clinical data. Therefore, the true number of infected or just colonized patients is not available. This should be mentioned in the discussion.

-          The huge difference in the incidence of clinical pneumonia in the 2 stages deserves a specific discussion. Particularly, it is important to specify whether the diagnosis criteria and the type (HAP vs VAPearly-late) have been changed in the 2 periods.

-          In general, I strongly suggest to shorten the discussion and to focus only on the points highlighted by the study. For instance, the discussion on the fungal infections seems to be beyond data presented.

Author Response

MAJOR

The author proposed an experience of antimicrobial stewardship ICU program (ASP) with relevant results in terms of antibiotic therapy prescription. The data are interesting but need some further clarification. First of all, the aim of the study should be defined and homogenized in the different sections of the manuscript. By results and discussion, the aim seems to be the evaluation of the effectiveness of the ASP in terms of antibiotic use and micro-organisms antibiotic resistance rather than a feasibility study. Unfortunately, the effects of ASP on infection incidence is unclear because data on pneumonia are incomplete. Moreover, differences in populations (e.g. SAPS II, infection at admission) and clinical management (e.g. infection control, surveillance cultures) of ICU patients, beyond ASP, between 2014 and 2016 might be significant and interfere with the results observed. More data on the population and the standard clinical practice in the 2 periods should be reported.

Response: Thanks for the reviewer’s comments. We have to agree that the manuscript was not adequately reported. We are pleased you consider the data of interest and we have made further clarifications. We have harmonized and clarified the main goal of the study. We agree with the confusion between feasibility and evaluation of effectiveness and we have tried to explain better that this study aim to provide a contemporary report of current clinical practice in ICU after a period of multidisciplinary AMS. We agree that the pneumonia data are incomplete and we have, following reviewer’s comments, included as a final major limitation that the criticism raised by the reviewer. We also included as a comment that the samples were taken when clinically indicated and there was no discrimination of timing to admission. No changes in sampling policies were implemented during the recent years. All the reviewer’s comments have been included as limitations.

MINOR

ABSTRACT

-          For antimicrobial stewardship, 2 acronyms (ASP and AMS) have been used. I suggest to use just one

Response: Done. We have chosen AMS as more commonly used in our settings.

-          The intervention (ASP) should be better clarified in terms of activities and time

Response: We have included in methods how this was implemented.

-          A conclusion should be added

Response: It has been added following reviewer’s comments.

INTRODUCTION

                Lines 536-538: The aim of the study should be better clarified: in the abstract and in the methods the aim seems to be the effectiveness of ASP, whereas in the title and in the introduction seems to be only feasibility.

Response: Reviewer is right. Our AMS program was introduced in 2014 and we wanted to analyze the feasibility of AMS in critical care settings. In other words, what we detected and if that impacted in daily clinical practice after a 2-year period in critical care settings. We have clarified in the introduction.

METHODS

-          Line 545: Sensitivity (to antibiotics) might be changed to susceptibility to antibiotics.

Response: Done thorough the whole document.

-          Line 565: Was tracheal aspirate not considered or never performed?

Response: Blood cultures and endotracheal Tracheal aspirates (ETA) were obtained when clinically infection was suspected. We have included this sentence in methods.

-          Line 566: please clarify ‘various fluid samples (categorised as pleural of various other drain samples)’.

Response: We apologize for this error. Fluids means (Either pleural or abdominal drain samples) and it has been clarified.

-          Lines 566-568: the authors stated that they did not consider urine, rectal swab, line tip and viral samples because the study was focused on bloodstreams and respiratory infections. various fluid samples (categorised as pleural of various other drain samples) and fecal samples. Therefore, it is unclear the reason for including fecal samples in the analysis

Response: We have corrected this mistake. We considered rectal swabs for infection control purposes as our CDI is almost zero for the last years.

-          Have a CFU threshold used in the evaluation of the respiratory tract samples ?

Response: CFU were not obtained in this study and we have added as a limitation.

-          Lines 572-577: I suggest rephrasing the primary end point. DOT could be not a endpoint, but just a variable. The reduction of DOT may be an end-point. The goal (antibiotic consumption improvement) is unclear. More, the relationship between end-point and hypothesis is difficult to realize. In general, the hypothesis should be placed at the beginning of the paragraph (or better in the introduction)

Response: We agree with all and it has been corrected accordingly. Moreover, hypothesis and goals have been deleted and placed in the introduction.

-          Lines 579-583: As above, the secondary end-points have to be better defined.

Response: Secondary endpoints have been modified accordingly to a reduction in the mentioned elements

RESULTS

-          Lines 632-633: The lines should be rephrased.

Response: The sentence have been rephrased

-          Lines 632-645: I suggest to move this section in the methods

Response : We have modified in methods and made more clear the text.

-          Lines 644-646: Is the difference in susceptibility significant ?

Response: We have only reported the DOT for antifungals without including susceptibility

-          Lines 654-663: I suggest to shorten these data and to use a table

Response: We agree and apologize for having repeated the data in text and tables. The text does now refer to table 2

-          Lines 664-669: The rate of candidemia should be reported for the 2 study periods and also the type of Candida. Data on respiratory isolation without data on invasive candida may be misleading.

Response:  Reviewer is right and there were no differences in candidemia as it is reported in table 1 and highlighted and added in the text.

-          The median (IQR) number of respiratory and blood samples for each patient should be reported. It is important to specify if the ASP changed the standard practice in terms of sampling (surveillance culture ?)

Response: Unfortunately we don’t have this data and represents a major limitation. We have included this as the first and most important limitation of our study and clarifying that standard practice in terms of sampling did not changed.

DISCUSSION

-          Lines 672-673: Again, the end-point is different to end-points defined in the abstract, introduction and methods

Response: We have merged the same goal in all the three parts and we deeply apologize and thanks the reviewer’s comment.

-          Limitations: The study is based on microbiological data and not on clinical data. Therefore, the true number of infected or just colonized patients is not available. This should be mentioned in the discussion.

Response: It is added in limitations literally as the reviewer correctly suggested with the following sentence: Finally, the true number of infected (HAP vs. VAP early/late ratio) or just colonized patients was not available. Because pneumonia diagnosis was mainly clinical, both diagnosis and the sampling were reduced dramatically that has impacted in episodes treated.

-          The huge difference in the incidence of clinical pneumonia in the 2 stages deserves a specific discussion. Particularly, it is important to specify whether the diagnosis criteria and the type (HAP vs VAPearly-late) have been changed in the 2 periods.

Response: We agree that this is an important comment. The episode of pneumonia was based on clinical criteria and antibiotics provided and in the second period, the sampling was reduced dramatically that has impacted in episodes treated. We have included this important remark in the limitation section. Please also see the response to the previous question as it is now included in limitations.    

-          In general, I strongly suggest to shorten the discussion and to focus only on the points highlighted by the study. For instance, the discussion on the fungal infections seems to be beyond data presented.

Response: We agree and the discussion has been shortened by 20% and the fungal infection section deleted.

Round 2

Reviewer 1 Report

I really think that the new version has improved the manuscript. However, I also think that a brief description of the general design of the study is not clearly exposed either in the abstract or in the methods section, and I suggest improving this point (sometimes it looks like the first audit included the analysis post intervention). The other point to be improved is the bibliography, as there are some errors in the way it has been presented and it should be reviewed. 

Finally, in the sentence “During the 6-months surveillance period, a daily multidisciplinary ward round including (Intensivists, clinical microbiologists and pharmacists)….”should say “During the 6-months surveillance period, a daily multidisciplinary ward round (including Intensivists, clinical microbiologists and pharmacists)….”

Author Response

I really think that the new version has improved the manuscript. However, I also think that a brief description of the general design of the study is not clearly exposed either in the abstract or in the methods section, and I suggest improving this point (sometimes it looks like the first audit included the analysis post intervention). The other point to be improved is the bibliography, as there are some errors in the way it has been presented and it should be reviewed. 

Response, we thank the reviewer for helping these authors to improve the manuscript. We agree with the comments and it has been clarified. We have also followed journal's recommendations for the references

Finally, in the sentence “During the 6-months surveillance period, a daily multidisciplinary ward round including (Intensivists, clinical microbiologists and pharmacists)….”should say “During the 6-months surveillance period, a daily multidisciplinary ward round (including Intensivists, clinical microbiologists and pharmacists)….”

Response: we agree and we have changed the bracket to the right place